# Highly Sensitive Voltammetric Glucose Biosensor Based on Glucose Oxidase Encapsulated in a Chitosan/Kappa-Carrageenan/Gold Nanoparticle Bionanocomposite

**DOI:** 10.3390/s19010154

**Published:** 2019-01-04

**Authors:** Ilhem Rassas, Mohamed Braiek, Anne Bonhomme, Francois Bessueille, Guy Raffin, Hatem Majdoub, Nicole Jaffrezic-Renault

**Affiliations:** 1Institute of Analytical Sciences, University of Lyon, 69100 Villeurbanne, France; ilhemras@hotmail.fr (I.R.); mohamed_braiek@yahoo.fr (M.B.); anne.bonhomme@isa-lyon.fr (A.B.); francois.bessueille@univ-lyon1.fr (F.B.); guy.raffin@isa-lyon.fr (G.R.); 2Laboratory of Advanced Materials and Interfaces, University of Monastir, 5000 Monastir, Tunisia; hatem.majdoub@fsm.rnu.tn

**Keywords:** chitosan, kappa-carrageenan, polyelectrolyte complex, gold nanoparticles encapsulation, glucose oxidase, bionanocomposite

## Abstract

In this work, an enzymatic sensor, based on a bionanocomposite film consisting of a polyelectrolyte complex (PEC) (Chitosan/kappa-carrageenan) doped with gold nanoparticles (AuNPs) encapsulating glucose oxidase (GOD) deposited on a gold electrode (Au) for glucose sensing, is described. Using the electrocatalytic synergy of AuNPs and GOD as a model of enzyme, the variation of the current (µA) as a function of the log of the glucose concentration (log [glucose]), shows three times higher sensitivity for the modified electrode (283.9) compared to that of the PEC/GOD modified electrode (93.7), with a detection limit of about 5 µM and a linearity range between 10 µM and 7 mM. The response of the PEC/AuNPs/GOD based biosensor also presents good reproducibility, stability, and negligible interfering effects from ascorbic acid, uric acid, urea, and creatinine. The applicability of the PEC/AuNPs/GOD based biosensor was tested in glucose-spiked saliva samples and acceptable recovery rates were obtained.

## 1. Introduction

Many polysaccharide-based polyelectrolyte complexes (PECs) have been used to make capsules to encapsulate drugs, in order to enhance their controlled and prolonged release. In particular, chitosan/kappa-carrageenan PEC capsules have been used for theophylline encapsulation. Its release rate has been shown to decrease when the concentration of chitosan increases, the capsule diameter increasing accordingly [1]. In the case of diltiazem encapsulation, better results have been obtained with chitosan/alginate PEC capsules, because carrageenan promotes the entry of water into the matrix, leading to capsule disintegration instead of swelling [2]. For sodium diclofenac, upon cross-linking with glutaraldehyde, prolonged drug release was obtained over 24 h [3]. Higher encapsulation efficiency of glucose oxidase (GOD) (isoelectric point 4.2) in chitosan/carrageenan complexes was observed as compared to that of a single polymer. The efficiency was doubled in polymer blends, which exhibited greater ability to encapsulate protein compared to that of a single polymer. At a charge ratio of 3, chitosan/kappa-carrageenan complexes showed the highest encapsulation efficiency (79%) [4]. The ability of chitosan-carrageenan PEC capsules to protect protein integrity under acidic conditions makes them a promising drug delivery system for the oral administration of peptides and proteins.

Chitosan-based electrochemical biosensors use chitosan as an electrode coating in order to limit interference, glucose oxidase being grafted on top of the chitosan layer [5,6]. Our recent paper reported the use of a chitosan-carrageenan polyelectrolyte in a voltammetric glucose biosensor. This easy-to-prepare biosensor exhibits a broad linear range, a low detection limit, excellent selectivity, and is shown to be applicable to saliva samples [7].

In recent years, considerable attention has been paid to improving the construction of biosensors using metal nanoparticles such as silver (Ag) [8], platinum (Pt) [9], and gold (Au) [10,11] because of their unique physical and chemical properties. In particular, gold nanoparticles have been used in the development of biosensor performance due to their ability to play an important role in immobilizing biomolecules while maintaining biocatalytic activity, their large specific surface area, their excellent biocompatibility and electrocatalytic properties [12,13].

Many types of electrochemical biosensor, such as immunological [14], the DNA [15] enzyme [16,17,18] sensors, with improved analytical performance, have been prepared on the basis of gold nanoparticles (AuNPs). The interaction of chitosan with gold nanoparticles has been studied; the stabilization of gold nanoparticles and the self-organization effect has been demonstrated [19,20]. In this work, the effect of AuNP doping of chitosan-carrageenan polyelectrolyte on the analytical performance of a voltammetric glucose biosensor is studied.

## 2. Materials and Methods

### 2.1. Reagents

Kappa-carrageenan is a natural polysaccharide extracted from red algae of the genus *Halurus* (MW = 750 kDa) with a polydispersity index of 1.7. Chitosan (average MW = 45 kDa with a degree of acetylation >75%) was obtained from Sigma-Aldrich. Citrate gold nanoparticles (AuNPs, 23 nm) were synthesized using the classic citrate reduction method [21]. Glutaraldehyde (GA, 25 wt% aqueous solution), 4-aminothiolphenol (4-ATP, 97%), and the enzyme glucose oxidase extracted from *Aspergillus niger* (GOD, 100–250 kU/g solid, type X-S) were purchased from Sigma-Aldrich (Saint-Quentin-Fallavier, France). *N*-(2-Hydroxyethyl) piperazine-*N*′-(2-ethanesulfonic acid) (HEPES), acetic acid (99.8%), and D-(+)-Glucose were also purchased from Sigma-Aldrich. The ethanol, acetone, sulfuric acid (95%), and hydrogen peroxide (30% in water) were purchased from Fluka and used for gold electrode pretreatment. Phosphate buffered saline (PBS, pH 7.4, 0.1 M) was prepared from monopotassic and dipotassic phosphate salts, sodium and potassium chlorides from Sigma-Aldrich and adjusted to pH 7.4. Potassium ferricyanide [K_3_Fe(CN)_6_] and potassium ferrocyanide [K_4_Fe(CN)_6_·3H_2_O] were provided by Sigma-Aldrich. All reagents were used without further purification. Aqueous solutions were prepared using ultrapure water from a MilliQ purification system.

### 2.2. Synthesis of Gold Nanoparticles

AuNPs were synthetized according to the classical following procedure [21]. Two hundred and fifty microliters of an aqueous solution of tetrachloroauric (III) acid trihydrate HAuCl_4_ 2% (*w*/*w*) was dissolved in 60 mL ultrapure water in a 150 mL beaker. The solution was heated and stirred until it boiled.

In addition, 1 mL of 1% (*w*/*w*) aqueous solution of trisodium citrate was prepared and added to the first beaker under vigorous stirring. The color of the gold chloride solution turned from pale yellow to wine-red 60 s after heating was stopped, indicating the formation of a gold cluster. The mixture vessel was then allowed to cool to room temperature. The final stabilized AuNPs were characterized by UV-vis spectrometry and the results are presented in Figure 1a. The absorption peak found at around 527 nm is attributed to the characteristic plasmon band of the AuNPs.

The size distribution of 23 ± 3 nm and the morphology of AuNPs were determined by transmission electron microscopy (TEM) as shown in Figure 1b. The transmission electron microscopy (TEM) images were obtained using a Philips CM120 with an accelerating voltage of 120 kV. AuNPs were examined after deposition of 3 μL of diluted solutions on a formvar-coated copper grid and evaporation to dryness.

### 2.3. Instrumentation

In this work, all electrochemical measurements were performed by cyclic voltammetry (CV) and square wave voltammetry (SWV) at room temperature using a Voltalab 80 model PGZ 402 analyzer instrument (Hach Lange, France) controlled with Voltamaster 4.0 software. All experiments were carried out using a 5 mL electrochemical cell connected to a conventional three electrode system with a platinum plate as an auxiliary electrode (active surface: 0.19 cm^2^), a saturated calomel electrode (SCE) as a reference electrode, and a gold plate electrode as the working electrode (active surface: 0.07 cm^2^).

Voltammetric experiments were carried out under stirring in 0.1 M phosphate buffer solution (PBS) pH 7.4 containing 10 mM Fe[(CN)]^3+^/^4+^ couple. The potential was cycled from 0 mV and +800 mV with a scan rate of 100 mV/s. Square wave voltammetry (SWV) was carried out between 0 mV and +800 mV with a pulse amplitude of 1 mV, a pulse width of 0.02 ms, and a potential increment of 10 mV.

AFM was used to investigate the surface topography of the PEC/AuNPs/GOD film immobilized on the clean gold surfaces. The AFM measurements were carried out using an Agilent 5500 AFM (Agilent Technologies, Palo Alto, CA, USA). Silicon tips with a nominal spring constant of 20 nm^−1^ were used in tapping mode at a frequency of ~300 kHz.

### 2.4. Preparation of Polysaccharide Complex PEC/AuNPs/GOD Mixture

The polyelectrolyte complex (PEC) with oppositely charged chitosan-carrageenan showed a high encapsulation efficiency of 79% at charge ratios of 3. The kappa-carrageenan solution (1 mg/2 mL of water) was mixed with a chitosan solution to complete the coacervation process. The chitosan was prepared with 3 mg in 2 mL of liquid composed of 1.4 mL of HEPES buffer solution at pH 7.4 completed with 0.6 mL of acetic acid. The mixture was allowed to stabilize at room temperature for 15 min. Then, for the encapsulation of the enzyme, 5 mg of glucose oxidase (GOD) was added to the 1 mL polyelectrolyte complex solution and mixed again. Another 15 min of stabilization at room temperature was then allowed. Twenty microliters of AuNPs (4 × 10^3^ AuNPs per µL) was added to 1 mL of the PEC/GOD and mixed again, followed by 15 min of stabilization. Thereafter the final mixture was kept for 24 h at 4 °C.

### 2.5. Elaboration of the Glucose Biosensor

Gold substrates were fabricated by the French RENATECH network (LAAS, CNRS Toulouse). They were constructed using standard silicon technologies. Silicon wafers ((100)-oriented, P-type (3–5 Ω.cm)) were thermally oxidized to grow an 800 nm-thick field oxide. Then, a 30 nm-thick titanium layer followed by a 300 nm-thick gold top layer were deposited by evaporation under vacuum.

Before functionalization of the working electrode, the surface of a gold microelectrode (1.2 × 1.2 cm^2^ square plates) was cleaned for 5 min in acetone in an ultrasonic bath, then in ultrapure water. Finally, it was dipped for 5 min at room temperature in a piranha solution (H_2_SO_4_:H_2_O_2_ = 3:1 *v*/*v*), then cleaned in ultrapure water, and dried under a nitrogen flow.

Subsequently, the gold electrode was incubated in an ethanolic solution of 4-ATP (10 mM) for 12 h at 4 °C in order to form an anchoring layer at the electrode surface. Then the substrate was rinsed with ethanol to remove the unbonded thiols and dried under nitrogen stream. After that, 10 µL of the homogenized mixture [PEC/AuNPs/GOD] was dropped onto the working electrode surface. The modified electrode was then placed for 20 min in saturated glutaraldehyde vapor (cross-linker) and was stored for 24 h at 4 °C. All the modified electrodes were immersed in PBS (0.1 M, pH 7.4) to examine their water stability. The different steps of preparation of the glucose biosensor are summarized in Figure 2.

The morphology of the PEC/AuNPs/GOD film deposited on the gold slides was exhibited using AFM (Figure 3). Some figures, due to stratification of the film, appear on the surface. This phenomenon is due to the interaction of PEC with gold nanoparticles, leading to the self-organization effect [19,20]. The RMS (root mean squared) roughness of the chitosan-carrageenan polyelectrolyte film is found to be 2.2 nm and is very close to that of the bare gold surface (RMS roughness 1.6 nm). No figure such as that presented in Figure 3 was observed without gold nanoparticles.

## 3. Results

### 3.1. Electrochemical Characterization of the Modified Gold Electrode

First, cyclic voltammetry (CV) was used to investigate the electrochemical properties of the gold electrodes modified by the complex of polysaccharides mixed with glucose oxidase (PEC/GOD/Au) before and after AuNP doping.

The modified gold electrode was electrochemically characterized in 0.1 M phosphate buffer solution at pH 7.4 in 1 mM of [Fe (CN)_6_]^3/4−^ solution. The separation between anodic and cathodic peak potentials (∆Ep = Epa − Epc) as well as the intensity of the peaks (Ipa and Ipc) are correlated to the electron transfer properties of the electrodes.

The cyclic voltammograms before and after doping with AuNPs are presented in Figure 4. Both oxidation and reduction peaks of the Fe(CN)_6_^3−^/Fe(CN)_6_^4−^ couple are clearly detected, the observed value of ∆Ep is 0.09 mV and anodic peaks 397 µA. After AuNPs doping (Figure 4b) of the biocomposite film (PEC/AuNPs/GOD), ∆Ep slightly increases from 0.09 mV to 0.12 mV and the anodic peak greatly increases from 397 µA to 996 µA, indicating an increase of the electron transfer rate in the presence of Au nanoparticles.

### 3.2. SWV Response of the Glucose Biosensor

The SWV response of the glucose biosensor was studied at a concentration of 0.2 mM of glucose. As seen in Figure 5, the SWV anodic peak obtained by the electrode modified by PEC/GOD/Au is (curve a in Figure 5) 460 mV and the peak intensity value is 111 µA.

After AuNPs doping of the biofilm, the intensity of the anodic peak (curve b in Figure 5) increases from 111 µA to 217 µA, and the peak potential value also shifts towards positive values, from 460 mV to 470 mV.

### 3.3. SWV Response of PEC/AuNPs/GOD/Au Biosensor when the Concentration of Glucose Increases

The electroanalytical characterization of glucose biosensors was performed by using SWV as the detection technique, recording the current intensity at the PEC/AuNPs/GOD/Au modified electrode at different concentrations of glucose after 5 min of contact. This time was sufficient to achieve a stable signal for each injected concentration. As seen in Figure 6, the biosensor signal increased with glucose concentration in the 0–7 mM range. The sensitivity of the technique, corresponding to the slope of the linear part of the curve, was 283.9 µA/log[glucose].

The calibration curve shown in Figure 7 presents a limit of detection (LOD), calculated as the concentration of glucose giving a signal equal to three times the standard deviation on the blank (in the absence of glucose) of 5 μM, and a dynamic linear range between 10 µM to 7 mM.

### 3.4. Analytical Performance of the Glucose Biosensor

#### 3.4.1. Comparison of Biosensor Response Recorded for Glucose in Absence and Presence of AuNPs

As shown in Figure 8, the maximum SWV response for 7 mM of glucose, obtained with a PEC/AuNPs/GOD modified electrode, is 791 µA, compared to 261 μA for the PEC/GOD modified one [7]. The presence of these gold nanoparticles improves the performance of the glucose sensor based on PEC/GOD, due to the electrocatalytic properties of AuNPs.

The Michaelis-Menten constant can be determined from the calibration curve presented in Figure 8. In presence of gold nanoparticles, the saturation is obtained for a concentration of glucose of 7 mM, then Vmax corresponds to a current of 791 µA. Vmax/2 corresponds to a current of 395 µM, then to a concentration of 10^−3.35^ M (5.6 × 10^−4^ M). The same value of Michaelis-Menten constant is obtained without gold nanoparticles. This low value of Km, compared to the Km of free enzyme (33–110 mM) shows a higher stability of the substrate-enzyme complex in the chitosan-carrageenan polyelectrolyte.

The analytical performance of the glucose biosensor based on PEC and glucose oxidase, with and without AuNP doping, is summarized in Table 1. The relative standard deviation is calculated from three different biosensors. The AuNP doping improves the sensitivity of glucose detection and inter-sensor reproducibility. The LOD value was calculated according to the following expression: 2σ/ISI, σ being the noise from the blank, and ISI the sensitivity, slope of the calibration curve. We obtained the same value 5 µM, with and without AuNPs, due to a higher noise of the blank.

As shown in Table 2, the analytical characteristics obtained in this study were compared with those of some typical glucose biosensors based on AuNPs and another type of polysaccharide. The results obtained are in the same range as the best voltammetric/amperometric glucose biosensors based on relevant AuNPs matrices with polysaccharides, some of them being limited in the dynamic range.

#### 3.4.2. Reproducibility and Shelf Life of the PEC/AuNPs/GOD Based Glucose Biosensor

Three independent electrodes were used to measure reproducibility at different glucose concentrations. The resulting relative standard deviation indicates good reproducibility sensor to sensor (4.5%).

Stability is also an important characteristic for the performance of a biosensor. This property was studied over three weeks by using one single glucose biosensor. The results show that the sensor remained stable by providing the same detection range and sensitivity during that period, and also a good relative standard deviation (RSD = 4.5%). The glucose biosensors based on the polyelectrolyte complex doped with AuNPs were stored at 4 °C in a refrigerator when not in use.

#### 3.4.3. Selectivity

In the presence of different analytes which are classical interfering species in glucose determination (uric acid (Figure 9a), ascorbic acid (Figure 9b), urea (Figure 9c), and creatinine (Figure 9d)), the PEC/AuNPs/GOD based biosensor was tested for the detection of 0.2 mM of glucose, to demonstrate the selectivity of this biosensor.

The percentage of variation of the peak maximum for the detection of 0.2 mM glucose concentration in the presence of the interfering substances is presented in Figure 9: 99.91% for uric acid, 99.78% for ascorbic acid, 100.17% for urea, and 99.66% for creatinine. These results show that the detection of glucose with PEC/AuNPs/GOD film limits the interference of the electroactive species.

#### 3.4.4. Real Sample Analysis

The applicability of the glucose biosensor based on the PEC/AuNPs/GOD modified gold electrode was investigated. The recovery of glucose was determined in spiked saliva samples after diluting it (10%) in PBS (pH 7.4).

As seen in Table 3, the proposed biosensor showed excellent reliability and accuracy for glucose detection in real samples.

## 4. Conclusions

This study describes a fast and easy way to elaborate chitosan/kappa-carrageenan (PEC)-gold nanoparticle films using SWV and CV electrochemical methods for the glucose biosensor response.

In addition, the resulting PEC film containing gold nanoparticles can be used to construct biosensors by assembling enzymes on the surface of the film. The immobilized enzymes retain good bioactivity (283.9 µA/log[glucose]) and stability. The obtained biosensor presents good selectivity and is applicable to real biological samples such as saliva and serum. The film fabrication technique demonstrated in this work and its application to biosensors is readily applicable to the fabrication of other PEC/metallic nanoparticle based enzymatic films for biosensors and biomaterials.

## Figures and Tables

**Figure 1 sensors-19-00154-f001:**
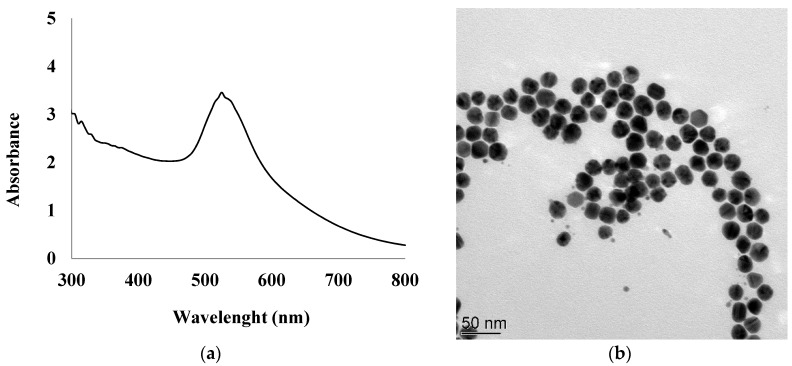
UV–vis spectrum of synthesized gold nanoparticles (AuNPs) suspension (**a**), TEM image of AuNPs (**b**).

**Figure 2 sensors-19-00154-f002:**
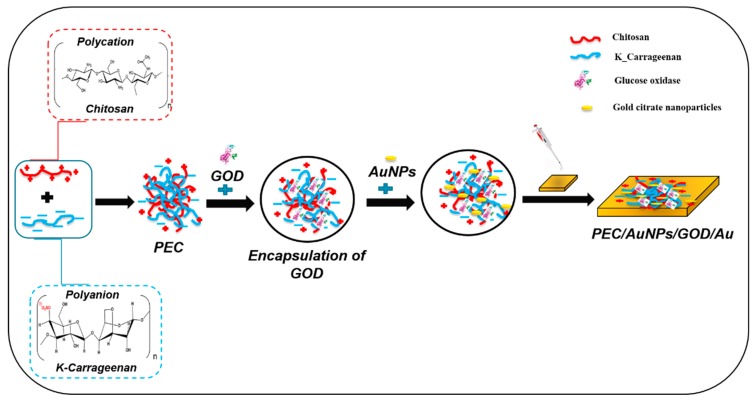
Schematic drawing of complexation of oppositely charged polysaccharides coupled with gold nanoparticles and elaboration of polyelectrolyte complex (PEC)/AuNPs/glucose oxidase (GOD)/Au.

**Figure 3 sensors-19-00154-f003:**
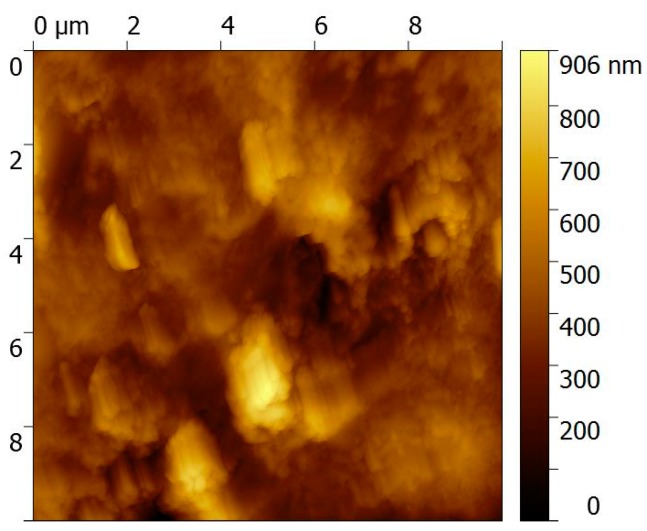
AFM image of a gold electrode modified with PEC/AuNPs/GOD/Au.

**Figure 4 sensors-19-00154-f004:**
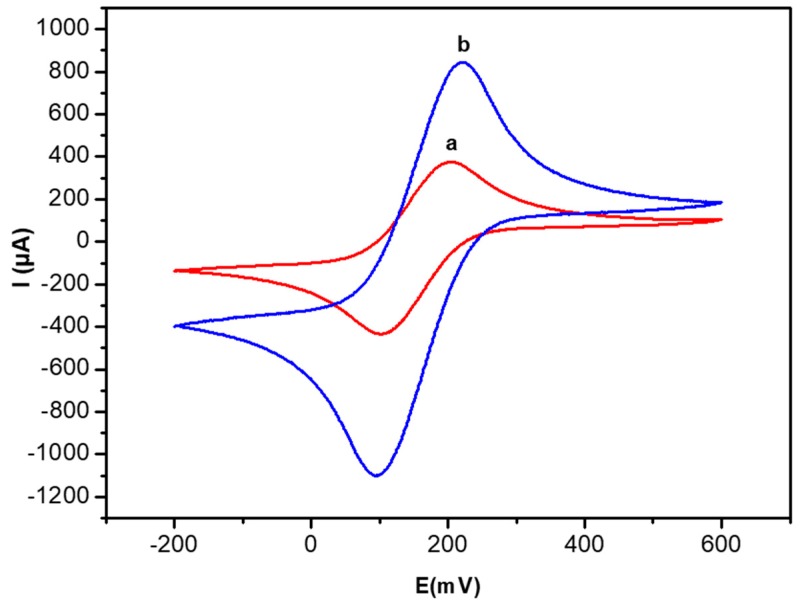
Cyclic voltammetry (CV) curves of the PEC/GOD modified gold electrode in the absence (**a**) and the presence of AuNPs (**b**). Measurements in 10 mM [Fe[(CN)_6_]^3−/4−^ PBS solution (0.1 M, pH 7.4) and scan rate 100 mV/s.

**Figure 5 sensors-19-00154-f005:**
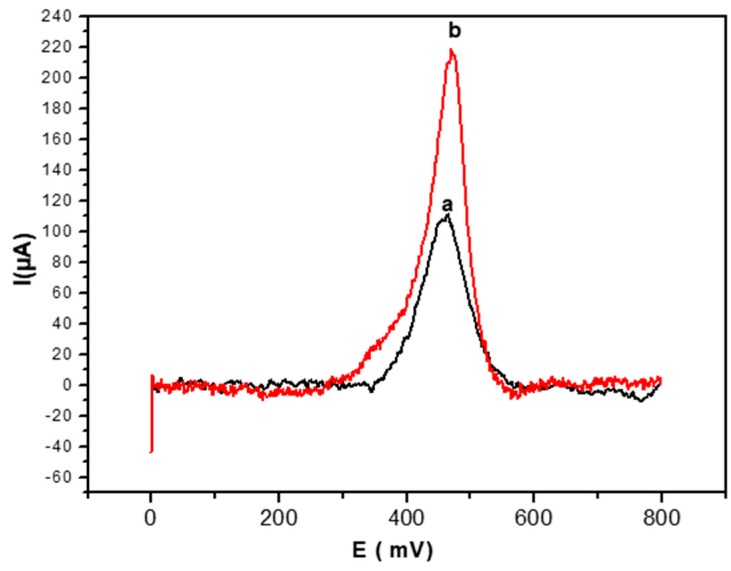
Square wave voltammetry (SWV) anodic peak for glucose detection (0.2 mM) obtained with the PEC/GOD modified gold electrode in the absence (**a**) and the presence (**b**) of AuNPs. SWV performed at 100 mV/s in PBS solution (0.1 M, pH 7.4).

**Figure 6 sensors-19-00154-f006:**
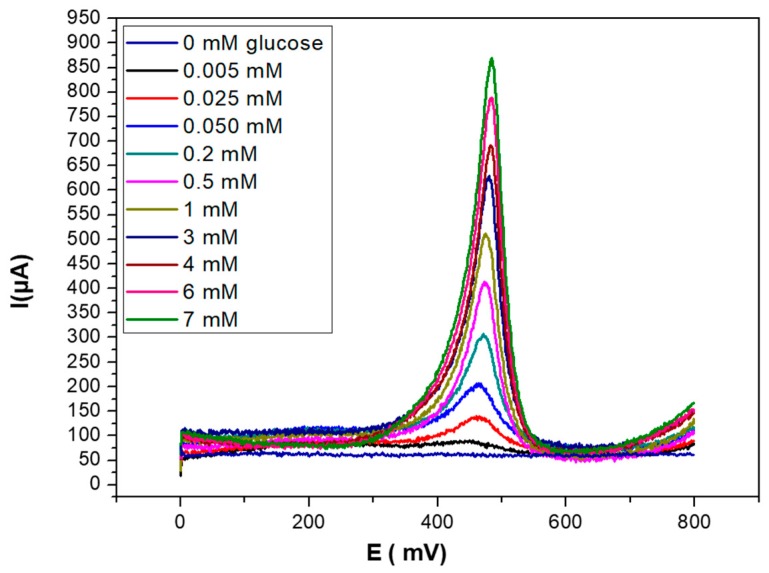
SWV curves obtained for PEC/AuNPs/GOD modified gold electrode upon injection of increasing concentrations of glucose (0–7 mM), in PBS solution (0.1 M, pH 7.4).

**Figure 7 sensors-19-00154-f007:**
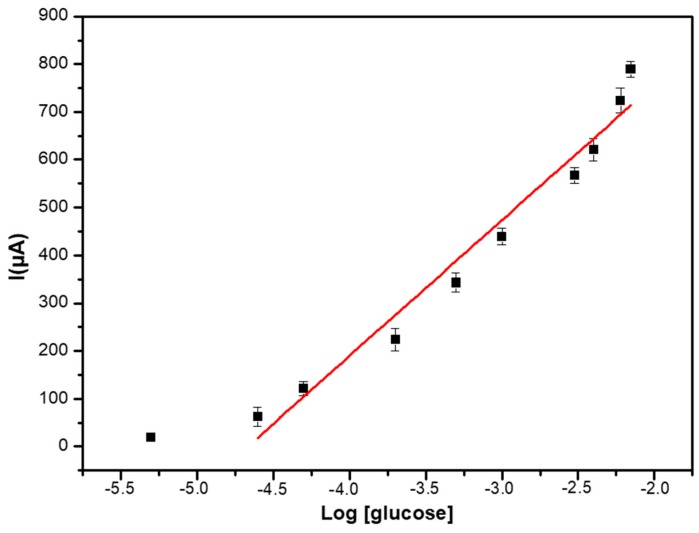
Calibration curve of the PEC/AuNPs/GOD/Au biosensor in PBS solution (0.1 M, pH 7.4).

**Figure 8 sensors-19-00154-f008:**
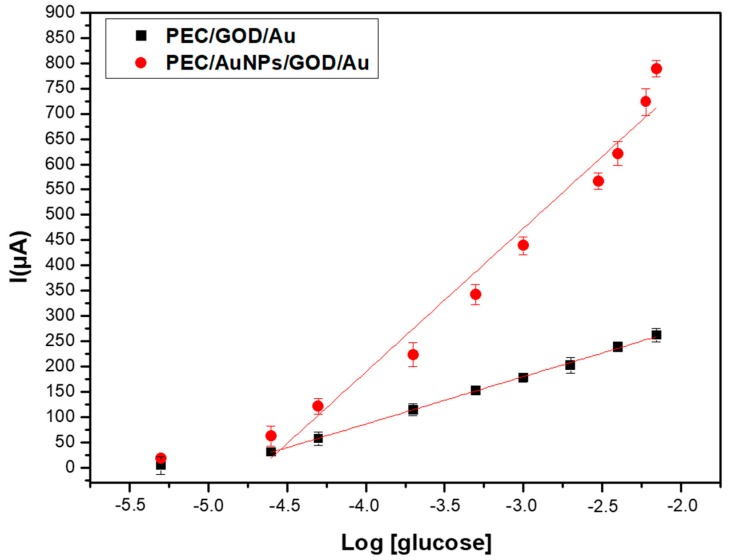
Comparison of the SWV responses of PEC/AuNPs/GOD/Au biosensor recorded for glucose (5 µM–7 mM) in the absence (**■**) and presence of AuNPs (**●**).

**Figure 9 sensors-19-00154-f009:**
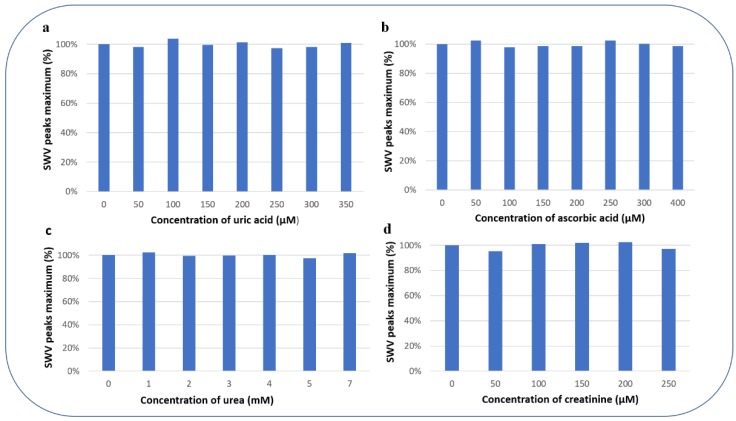
SWV peak maximum (%) for the detection of 0.2 mM glucose concentration in presence of different concentrations of possibly interfering substances (**a**) uric acid, (**b**) ascorbic acid, (**c**) urea, and (**d**) creatinine.

**Table 1 sensors-19-00154-t001:** Analytical performance of the glucose biosensor based PEC/GOD, with and without AuNP doping.

Biosensor	LOD for Glucose	Sensitivity µA/log[glucose]	RSD
**PEC/GOD/Au** (Biosensor without AuNPs)	5 µM	93.7	6%
**PEC/AuNPs/GOD/Au** (Biosensor with AuNPs)	5 µM	283.9	4.5%

**Table 2 sensors-19-00154-t002:** Comparison of the proposed biosensor with voltammetric/amperometric glucose biosensors based on GOD and polysaccharides coupled with AuNPs.

	Detection Limit	Dynamic Range	References
PEC/AuNPs/GOD/Au	5 µM	10 µM–7 mM	This work
Chitosan/PPy/AuNPs/ITO	3.1 µM	0–3.2 mM	[22]
Chitosan-GOD/AuNPs/GC	13 µM	50 µM–1.3 mM	[23]
Au/dithiol/Au/cystamine/GOx	8.2 µM	20 µM–5.7 mM	[24]
Amperometric detection (GOx-AuNP)_n_/Au	2.7 mM	0.005–2.4 mM	[25]
CS/β-G-GOD/AuNPs–CS/PB–CS/Au	1.56 μM	6.25–93.75 μM	[26]
(GCE/CHI–AuNPs)	370 µM	400 µL to 10.7 mM	[27]

**Table 3 sensors-19-00154-t003:** Glucose detection in real spiked saliva samples.

Added Concentration of Glucose	Recovery %
50 µM	106
0.2 mM	97

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
