# Peer review of "Highly Sensitive Voltammetric Glucose Biosensor Based on Glucose Oxidase Encapsulated in a Chitosan/Kappa-Carrageenan/Gold Nanoparticle Bionanocomposite"

_sensors, 2019, doi:10.3390/s19010154_

Round 1
Reviewer 1 Report
The presented work is interesting but needs further improvements for the readers' convenience.
The authors are kindly requested to provide more information about the influence of the buffer constituents on the biosensor performance since the mixture was prepared in HEPES buffer while the experiments were performed in PBS buffer.
Can the authors provide data on the Michaelis-Menten constants? How do they relate in the case of the considered biosensors?
Can the authors provide AFM images for the unmodified gold electrode or for the biosensor surface without NP?
The authors are kindly requested to also consider other recent bibliographic references.
Author Response
The authors thank reviewer #1 for valuable comments that will improve the quality of the manuscript.
Reviewer #1
The presented work is interesting but needs further improvements for the readers' convenience.
The authors are kindly requested to provide more information about the influence of the buffer constituents on the biosensor performance since the mixture was prepared in HEPES buffer while the experiments were performed in PBS buffer.
HEPES buffer is composed of an amphoteric molecule and is used in the formulation of the enzymatic mixture, in order to protect enzyme. After encapsulation in the chitosan-carrageenan polyelectrolyte, the enzyme is protected and electrochemical measurements can be performed in a more conductive buffer, phosphate buffer.
Can the authors provide data on the Michaelis-Menten constants? How do they relate in the case of the considered biosensors?
The Michaelis-Menten constant can be determined from the calibration curve presented in Figure 8. In presence of gold nanoparticles, the saturation is obtained for concentration of glucose of 7 mM, then Vmax correspond to a current of 791 µA. Vmax/2 corresponds to a current of 395 µM, then to a concentration of 10-3.35 M (5.6x10-4 M). The same value of Michaelis-Menten constant is obtained whithout gold nanoparticles. This low value of Km, compared to the Km of free enzyme (33-110 mM) shows a higher stability of the substrate-enzyme complex in the chitosan-carrageenan polyelectrolyte.
This point was reported in page 8.
Can the authors provide AFM images for the unmodified gold electrode or for the biosensor surface without NP?
The RMS roughness of the chitosan-carrageenan polyelectrolyte film is found to be 2.2 nm and is very close to that of the bare gold surface (RMS roughness 1.6 nm). No figure such as that presented in Figure 3 were observed without gold nanoparticles.
This point was reported in page 5.
The authors are kindly requested to also consider other recent bibliographic references. Two references were added in the introduction [12,13]
Reviewer 2 Report
The manuscript reported a highly sensitive voltammetric glucose biosensor based on glucose oxidase encapsulated in a chitosan/kappa-carrageenan/gold nanoparticle bionanocomposite. The proposed sensor showed a good sensitivity, dynamic range, selectivity, and applicable to real biological samples. Therefore, I recommend its accept for publication after the minor revision. The following are the specific comments:
1) Introduction. Nanomaterials, especially gold nanomaterials have been widely used for the fabrication of biosensors due to their inherent properties. Some representative papers in this field should be added and discussed (Analytical chemistry 2011, 83 (20), 7902-7909; Analytical chemistry 2009, 81 (16), 6641-6648)
2) Materials and Methods: The size distribution of 23 nm AuNPs should be provided, as in the Figure 1b.
3) The concentration of the AuNPs for the fabrication of biosensors should be characterized and provided.
4) For Figure 3, control experiments for the AFM images of PEC/AuNPs/Au should be provided, if possible.
5) In table 1, the LOD for the biosensors with and without AuNPs are 5 uM. Please check the value, since the authors claimed that the doping of AuNPs improved the sensitivity.
6) Conclusion: the potential application of the proposed sensor should be discussed.
Author Response
The authors thank reviewer #2 for valuable comments that will improve the quality of the manuscript.
Reviewer #2
The manuscript reported a highly sensitive voltammetric glucose biosensor based on glucose oxidase encapsulated in a chitosan/kappa-carrageenan/gold nanoparticle bionanocomposite. The proposed sensor showed a good sensitivity, dynamic range, selectivity, and applicable to real biological samples. Therefore, I recommend its accept for publication after the minor revision. The following are the specific comments:
1) Introduction. Nanomaterials, especially gold nanomaterials have been widely used for the fabrication of biosensors due to their inherent properties. Some representative papers in this field should be added and discussed (Analytical chemistry 2011, 83 (20), 7902-7909; Analytical chemistry 2009, 81 (16), 6641-6648)
Both references were added : Refs12 and 13
2) Materials and Methods: The size distribution of 23 nm AuNPs should be provided, as in the Figure 1b.
Size distribution of AuNPs was calculated from Figure 1b. The size distribution is of 23 ± 3 nm. This point is reported in page 3.
3) The concentration of the AuNPs for the fabrication of biosensors should be characterized and provided.
Concentration of AuNPs was determined through the intensity of the absorbance peak. It is equal to 4x103 AuNPs per µL. This value is reported in page 4.
4) For Figure 3, control experiments for the AFM images of PEC/AuNPs/Au should be provided, if possible.
The RMS roughness of the chitosan-carrageenan polyelectrolyte film is found to be 2.2 nm and is very close to that of the bare gold surface (RMS roughness 1.6 nm). No figure such as that presented in Figure 3 were observed without gold nanoparticles.
This point was reported in page 5.
5) In table 1, the LOD for the biosensors with and without AuNPs are 5 uM. Please check the value, since the authors claimed that the doping of AuNPs improved the sensitivity.
The LOD value was calculated according to the following expression: 2σ/ISI, σ being the noise from the blank, and ISI the sensitivity, slope of the calibration curve. We obtained the same value 5 µM, with and without AuNPs, due to a higher noise of the blank. This point was reported in page 9.
6) 
Conclusion: the potential application of the proposed sensor should be discussed.
The conclusion was completed as :
The obtained biosensor presents good selectivity and is applicable to real biological samples such as saliva and serum. The film fabrication technique demonstrated in this work and its application to biosensors is readily applicable to the fabrication of other PEC/metallic nanoparticle based enzymatic films for biosensors and biomaterials.
Round 2
Reviewer 1 Report
The authors provided responses to all recommendations, and the paper can be published.